

# Antimicrobial resistance and genetic relationships of enterococci from siblings and non-siblings *Heliconius erato phyllis* caterpillars

Rosana Huff[1], Rebeca Inhoque Pereira[2], Caroline Pissetti[3], Aldo Mellender de Araújo[4], Pedro Alves d'Azevedo[2], Jeverson Frazzon[5] and Ana Paula Guedes Frazzon[1]

[1] Institute of Basic Health Sciences, Department of Microbiology, Immunology and Parasitology, Federal University of Rio Grande do Sul, Porto Alegre, Rio Grande do Sul, Brazil
[2] Basic Health Sciences, Department of Microbiology, Health Sciences Federal University, Porto Alegre, Rio Grande do Sul, Brazil
[3] Department of Veterinary Preventive Medicine, Federal University of Rio Grande do Sul, Porto Alegre, Rio Grande do Sul, Brazil
[4] Institute of Biosciences, Genetic Department, Federal University of Rio Grande do Sul, Porto Alegre, Rio Grande do Sul, Brazil
[5] Food Science Institute, Federal University of Rio Grande do Sul, Porto Alegre, Rio Grande do Sul, Brazil

Corresponding author
Ana Paula Guedes
Frazzon, ana.frazzon@ufrgs.br

## ABSTRACT

**Background**. Studies evaluating bacteria in insects can provide information about host–microorganism–environment interactions. The gut microbial community has a profound effect on different physiological functions of insects. *Enterococcus* spp. are part of the gut community in humans and other animals, as well as in insects. The presence and antimicrobial resistance profile of enterococci are well studied in different animals; however, data for *Heliconius erato phyllis* (Lepidoptera: Nymphalidae) do not yet exist. Therefore, the aims of this study were to evaluate the distribution of enterococcal species, their antimicrobial resistance profile and virulence genes, and the genetic relationships between enterococci isolated from fecal samples from sibling and non-sibling *H. erato phyllis* caterpillars collected from different sites in South Brazil.

**Methods**. Three *H. erato phyllis* females were captured (two from a forest fragment and one from an urban area), and kept individually in open-air insectaries. Eggs were collected and caterpillars (siblings and non-siblings) were fed daily with *Passiflora suberosa* leaves. Fecal samples ($n = 12$) were collected from fifth-instar caterpillars, inoculated in selective medium, and 15 bacterial colonies were randomly selected from each sample. Enterococci were identified by PCR and MALDI-TOF, analyzed by disk diffusion antimicrobial susceptibility tests, and screened for resistance and virulence genes by PCR. The genetic relationships between the strains were determined using pulsed-field gel electrophoresis (PFGE).

**Results**. A total of 178 enterococci strains were identified: *E. casseliflavus* (74.15%; $n = 132$), *E. mundtii* (21.34%; $n = 38$), *E. faecalis* (1.12%; $n = 2$) and *Enterococcus* sp. (3.37%; $n = 6$). High rates of resistance to rifampicin (56%) and erythromycin (31%) were observed; 120 (67.41%) of the isolates showed resistance to at least one antibiotic and six (3.37%) were multidrug-resistant.None of the erythromycin-resistant strains was positive for the *erm*(B) and *msr*C genes. The virulence genes *esp*, *ace*, and *gelE*

were observed in 35%, 7%, and 1% of the strains, respectively. PFGE separated the enterococci into 22 patterns, four being composed of strains from sibling caterpillars. **Conclusion**. *Enterococcus casseliflavus* was the dominant species in fecal samples of fifth-instar caterpillars. Resistant enterococci strains may be related to environmental pollution or the resistome. The PFGE analysis showed genetic relationships between some strains, suggesting that the enterococci isolated from fecal samples of the sibling caterpillars might have come from common sources, e.g., via diet (herbivory) and/or vertical transmission (through the egg surface). Further studies will be conducted to better understand the role of *Enterococcus* in the microbial community of the gastrointestinal tract of these insects, and the mechanisms involved in acquisition and maintenance of enterococci.

## INTRODUCTION

*Heliconius* (Lepidoptera: Nymphalidae) represents a widespread genus of butterflies distributed throughout tropical and subtropical regions, from South America to the southern United States (*Brown, 1981*; *Merril et al., 2015*). Adults feed on pollen as well as nectar (*Gilbert, 1972*; *Merril et al., 2015*), and this food supply provides a better use of resources and a reduction in competition since the same flower can provide nutrients for both *Heliconius* and other butterflies that exclusively use nectar as food. This diet is rich in amino acids that allow adult females to oviposit on a daily basis during their lives (*Gilbert, 1972*). *Heliconius erato phyllis* is a subspecies commonly found in forests and urban environments from northeastern Brazil to northern Argentina. Adult females are monandrous, i.e., they mate only once, and lay their eggs individually in the apical meristem of a host plant, to minimize potential cannibalization of eggs by first-instar caterpillars (*De Nardin, Da Silva & Araújo, 2016*). Several species of passionflower vines, including *Passiflora suberosa*, *P. misera* and *P. capsularis*, are host plants for oviposition and feeding of caterpillars. These *Passiflora* species, commonly found in Southern Brazil where this study was conducted, have cyanogenic glycosides, which are assimilated by the caterpillars to make them unpalatable to potential predators. However, adults of *H. erato phyllis* are also able to synthesize cyanogenic glycosides and transfer them to their eggs (*Hay-Roe & Nation, 2007*).

It has been recognized for a long time that microorganisms play key roles in various physiological functions of animal hosts. The gut microbial community promotes an especially diverse range of benefits for insects, e.g., by improving nutrition via synthesis of vitamins and/or establishment of metabolic pathways, actively participating in degradation of xenobiotic compounds, and contributing to the defense against invading pathogens and immune system modulation (*Douglas, 2015*; *Shao et al., 2017*). The presence of microorganisms in the gastrointestinal (GI) tract of insects can be explained by environmental bacteria ingested with food and/or acquired by maternal transfer (*Engel*

& *Moran, 2013*). A growing number of studies have addressed the importance of the microbiota in the GI tract of insects (*Engel & Moran, 2013*; *Chen et al., 2016*; *Douglas, 2018*), and *Enterococcus* is one of the most frequent bacterial genera present in the gut microbiota at different life stages of Lepidoptera (*Brinkmann, Martens & Tebbe, 2008*; *Chen et al., 2016*; *Snyman et al., 2016*; van Shooten et al. 2018; *Allonsius et al., 2019*).

The *Enterococcus* genus is often found in the GI tract of humans and animals, as well as in the guts of insects (*Shao et al., 2017*). *Hammer, McMillan & Fierer (2014)* reported that *Enterococcus* was the most abundant genus found in immature stages and adults of *H. erato* from Panama. Furthermore, it has also been reported in insects of other orders, such as Coleoptera (*Kim et al., 2017*), Hymenoptera (*Audisio et al., 2011*), and Diptera (*Ghosh et al., 2014*). A characteristic of this genus is its intrinsic resistance to several antimicrobial agents and a great ability to transfer and acquire resistant genes (*Hollenbeck & Rice, 2012*). Despite the environmental resistome, the intense use of antimicrobials and anthropogenic activities, such as animal husbandry, agronomic practices, and wastewater treatment, play an important role in the emergence and spread of resistant-enterococci and/or antibiotic resistance genes in the environment, especially in soil, water, wastewater, and food (*Gothwal & Shashidhar, 2014*; *Singer et al., 2016*).

Antimicrobial resistance is one of the most serious public health problems, because of the spread of resistant bacteria leading to persistent infections, which are difficult to treat, and contamination of natural environments (*Watkins & Bonomo, 2016*; *Ferri et al., 2017*; *Aslam et al., 2018*). Insects have a wide distribution and can move freely between different environments; they may play an important role as reservoirs of drug-resistant strains and as their disseminators between animals and humans, especially when in contact with organic waste, livestock and their surrounding environment, and hospital facilities (*Zurek & Ghosh, 2014*; *Mohammed et al., 2016*; *Schaumburg et al., 2016*; *Zhang et al., 2017*; *Onwugamba et al., 2018*). In relation to insects caring antibiotic–resistant bacterial strains, studies have identified flies (*Ahmad et al., 2011*; *Usui et al., 2015*; *Mohammed et al., 2016*; *Schaumburg et al., 2016*; *Zhang et al., 2017*; *Onwugamba et al., 2018*) and cockroaches (*Ahmad et al., 2011*; *Pai, 2013*; *Moges et al., 2016*) as hosts of extended-spectrum beta-lactamase- and carbapenemase-producing Enterobacteriaceae, vancomycin-resistant *E. faecium* (VRE), and methicillin-resistant *Staphylococcus aureus* (MRSA). Insects collected from food establishments and in association with stored products were also found to be colonized by antimicrobial-resistant bacteria (*Macovei & Zurek, 2006*; *Channaiah et al., 2010*; *Mohammed et al., 2016*). Despite their importance, few studies have addressed the concern of insects carrying resistant enterococci (*Allen et al., 2009*; *Channaiah et al., 2010*; *Ahmad et al., 2011*; *Lowe & Romney, 2011*).

As previously mentioned, the *Enterococcus* genus is often found in the GI tract in Lepidoptera (*Hammer, McMillan & Fierer, 2014*; *Chen et al., 2016*; *Snyman et al., 2016*; van Shooten et al. 2018; *Allonsius et al., 2019*), and it may play a fundamental role in the development and regulation of bacterial communities in these insects (*Chen et al., 2016*; *Shao et al., 2017*). The identification of enterococcal strains and their resistance profile in insects is an important aspect that must be addressed for host–microorganism–environment interactions. To our knowledge, there have been no studies to date evaluating

enterococci in *H. erato phyllis*. The aims of our research were to analyze enterococcal species distribution, their antimicrobial resistance profile and virulence genes, and the genetic relationships between enterococci isolated from fecal samples from sibling and non-sibling *H. erato phyllis* caterpillars collected from different sites in South Brazil.

## MATERIALS & METHODS

### Sample collection

The fecal samples used in the present study were collected from fifth instar caterpillars. The caterpillars were sourced from three different populations of *Heliconius erato phyllis* butterflies and consisted of sibling from the same female. The *H. erato phyllis* females (HE) were captured with entomological nets in Rio Grande do Sul, South Brazil. The first female (HEAB2) was collected in a forest fragment located in Águas Belas Agronomical Station (30°02′18.1″S; 51°01′23.0″W), the second female (HEV2) from a population in an intense urban area in Viamão (30°09′40.5″S; 50°55′01.5″W) and the third female (HES2) in a forest fragment located in São Francisco de Paula (29°26′34.1″S; 50°36′48.8″W).

Butterflies were kept individually in open-air insectaries with dimensions of 2.3 m × 3 m × 3 m (width, length, height) approximately. Insectaries had many plants for simulation of natural conditions, including *P. suberosa*, used by females for oviposition. The butterflies were fed daily with a mixture containing water, honey and pollen.

A total of 12 eggs were collected (five from HEAB2, five from HEV2 and two from HES2) with the assistance of a paintbrush. The eggs were transported to the laboratory, and caterpillars were grown individually in cylindrical plastic pots. Immatures were fed exclusively with *P. suberosa* leaves (Fig. 1). Fecal samples were collected from each caterpillar individually after 48 h of molting to the fifth instar ($n = 12$), with the aid of a disposable plastic spoon, stored in 1.5 mL microtubes and maintained at −80 °C until processing. The oviposition dates are shown in Table S1.

This study was carried out in accordance with the recommendations of Chico Mendes Institute for Biodiversity Conservation (ICMBio). The protocol was approved by Information Authorization System in Biodiversity (SISBIO) number 33404-1. This study has the Council for the Management of Genetic Patrimony - CGEN - under the Ministry of Environment number A720680.

### Isolation and identification of Enterococci

Isolation and identification of enterococci were performed as previously described in *Santestevan et al. (2015)*, with modifications. One milligram of fecal sample was transferred to 10 mL of saline 0.85% and incubated at 37 °C for 24 h. One mL was inoculated in nine mL of Azide Dextrose Broth selective medium (Himedia, Mumbai, India) and incubated for 24 h at 37 °C. Aliquots (one mL) were placed in nine mL of saline 0.85%, and initial samples were further diluted 10-fold. From dilution $10^{-5}$ and $10^{-6}$, 100 µL was inoculated in brain heart infusion (BHI) agar plates (Himedia, Mumbai, India) supplemented with 6.5% NaCl, incubated for 48 h at 37 °C.

Fifteen colonies were randomly selected from each fecal sample. Phenotypic criteria (size/volume, shape, color, gram staining, catalase production and bile aesculin reaction)

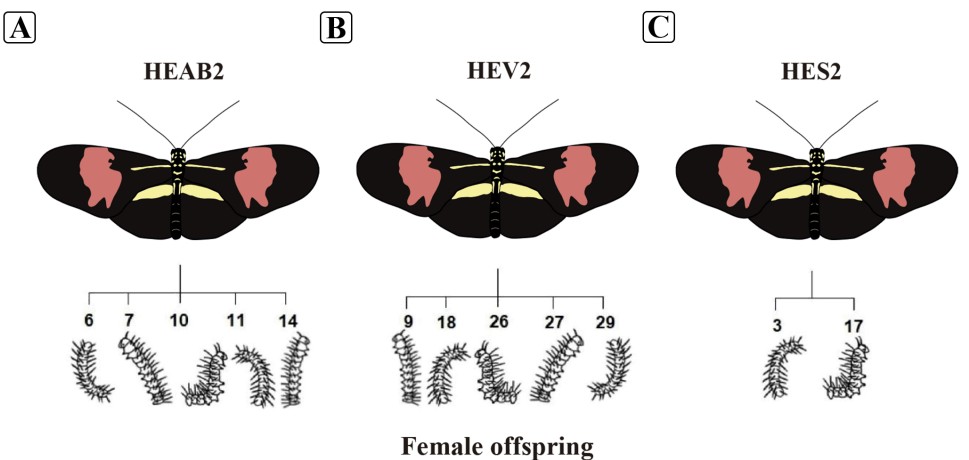

**Figure 1** **Illustration of *Heliconius erato phyllis* butterflies and caterpillars used in the study.** (A) Female HEAB2 and offspring (6, 7, 10, 11, and 14); (B) female HEV2 and offspring (9, 18, 26, 27, and 29); (C) female HES2 and offspring (3 and 17). Illustration credit: Lucas de Oliveira Einsfeld and Gabriella Oliveira de Araújo.

were used to separate the enterococci group and the non-enterococcal strains. Selected pure colonies were stored at −20 °C in a 10% (w/v) solution of skim milk (Difco, Sparks, MD, USA) and 10% (v/v) glycerol (Neon Comercial Ltda, São Paulo, SP, BR).

Genomic DNA was extracted by physicochemical method as previously described by *Depardieu, Perichon & Courvalin (2004)*. Polymerase chain reaction (PCR) assay was carried out using genus-specific primers targeting the *tuf* gene, which encodes elongation factor EF-Tu (*Ke et al., 1999*) (Table 1). Amplifications were performed in a total volume of 25 µL containing: 100 ng of template DNA, 1 X reaction buffer (Ludwig Biotechnology), 0.4 µM of each primer (Ludwig Biotechnology), 1.5 mM MgCl$_2$, 200 µM of dNTP (Ludwig Biotechnology), 1 U *Taq* DNA polymerase (Ludwig Biotechnology), and MilliQ water. Amplification was carried out in a conventional thermocycler (Applied Biosystems 2720 Thermal Cycler) according to the following program: initial denaturation at 95 °C for 3 min, followed by 35 cycles of 95 °C for 30 s, 54 °C for 30 s and 72 °C for 1 min, and a final extension at 72 °C for 7 min. The PCR products were visualized using electrophoresis on 1.5% (w/v) agarose gel, stained with SYBR® Safe DNA Gel, and visualized on a photocumenter. *E. faecalis* ATCC 29212 was used as positive control.

## Characterization of enterococci species

Isolates were screened with the species-specific PCR assay for *Enterococcus faecalis*, *Enterococcus faecium*, *Enterococcus casseliflavus* and *Enterococcus mundtii* (*Cheng et al., 1997*; *Jackson, Fedorka-Cray & Barrett, 2004*; *Sedgley et al., 2005*). The primers and annealing temperature used are listed in Table 1. Amplifications were prepared as described to *tuf* gene. PCR amplification was performed in the conventional thermocycler (Applied Biosystems 2720 Thermal Cycler) according to the following program: 94 °C for 5 min followed by 35 cycles of 94 °C for 1 min, appropriate annealing temperature for each species for 1 min, extension at 72 °C for 1 min, and a final extension at 72 °C for 5 min.
**Table 1  Primers used in the PCRs carried out in this study.**

| Primer | Nucleotide sequence (5′-3′) | AT[a] (°C) | Amplicon (bp) | Reference |
|---|---|---|---|---|
| Genus | | | | |
| tuf-F | TACTGACAAACCATTCATGATG | 54 | 112 | *Ke et al. (1999)* |
| tuf-R | AACTTCGTCACCAACGCGAAC | | | |
| *E. faecalis* | | | | |
| E16s-F | CCGAGTGCTTGCACTCAATTGG | 66 | 138 | *Sedgley et al. (2005)* |
| E16s-R | CTCTTATGCCATGCGGCATAAAC | | | |
| *E. faecium* | | | | |
| EM1A-F | TTGAGGGACACCAGATTGACG | 62 | 658 | *Cheng et al. (1997)* |
| EM1B-R | TATGACAGCGACTCCGATTCC | | | |
| *E. casseliflavus* | | | | |
| CA1 | TCCTGAATTAGGTGAAAAAAC | 59 | 288 | *Jackson, Fedorka-Cray & Barrett (2004)* |
| CA2 | GCTAGTTTACCGTCTTTAACG | | | |
| *E. mundtii* | | | | |
| MU1-F | CAGACATGGATGCTATTCCATCT | 60 | 94 | *Jackson, Fedorka-Cray & Barrett (2004)* |
| MU2-R | GCCATGATTTTCCAGAAGAATG | | | |
| *16s rRNA* | | | | |
| 8F | AGAGTTTGATCCTGGCTCAG | | | |
| 519F | CAGCAGCCGCGGTAATAC | 60 | 1,514 | *Coenye et al. (1999)* |
| 926R | CCGTCAATTCCTTTGAGTT | | | |
| 1522R | AAGGAGGTGATCCAGCCGCA | | | |
| Erythromycin | | | | |
| erm(B)_F | GAAAAGGTACTCAACCAAATA | 52 | 639 | *Sutcliffe, Tait-Kamradt & Wondrack (1996)* |
| erm(B)_R | AGTAACGGTACTTAAATTGTTTAC | | | |
| msrC_3 | AAGGAATCCTTCTCTCTCCG | 52 | 343 | *Werner, Hildebrandt & Witte (2001)* |
| msrC_4 | GTAAACAAAATCGTTCCCG | | | |
| Gelatinase | | | | |
| gelE_TE9 | ACCCCGTATCATTGGTTT | 50 | 419 | *Eaton & Gasson (2001)* |
| gelE_TE10 | ACGCATTGCTTTTCCATC | | | |
| Cytolysin | | | | |
| cylA_TE17 | TGGATGATAGTGATAGGAAGT | 59 | 517 | *Eaton & Gasson (2001)* |
| cylA_TE18 | TCTACAGTAAATCTTTCGTCA | | | |
| Adhesion | | | | |
| ace1_F | AAAGTAGAATTAGATCCACAC | 59 | 320 | *Mannu et al. (2003)* |
| ace2_R | TCTATCACATTCGGTTGCG | | | |
| Biofilm | | | | |
| ESP46 | TTACCAAGATGGTTCTGTAGGCAC | 60 | 913 | *Shankar et al. (1999)* |
| ESP47 | CCAAGTATACTTAGCATCTTTTGG | | | |
| Aggregation | | | | |
| agg_TE3 | AAGAAAAAGAAGTAGACCAAC | 60 | 1,553 | *Eaton & Gasson (2001)* |
| agg_TE4 | AAACGGCAAGACAAGTAAATA | | | |

**Notes.**
[a]AT, annealing temperature.

The DNA fragment amplified by PCR was analyzed in 1.5% (w/v) agarose gels stained with SYBR® Safe DNA Gel, and visualized on a photocumenter.

Strains that were not identified by PCR reactions were submitted to matrix-assisted laser desorption and ionization time-of-flight technique (MALDI-TOF) applied to *Enterococcus* sp., according to *Sauget et al. (2017)*.

Isolates classified as *Enterococcus* sp. were identified by Sanger sequence analysis. The PCR product of *16S rRNA* gene, using the 8F and R1522 primers (*Coenye et al., 1999*) (Table 1), was purified with Illustra™ GFX™ PCR DNA and gel band purification kit (GE Healthcare, Buckinghamshire, UK). To perform Sanger sequencing two additional primers, 519F and 926R, were used (*Coenye et al., 1999*) (Table 1). Sequencing was performed with the ABI PRISM® BigDye® Primer Cycle Sequencing Ready Reaction Kit in an ABI PRISM® 3100 Genetic Analyzer (Applied Biosystems®), according to the manufacturer's protocol. The sequence obtained was compared to nucleotide sequences of reference enterococci strains deposited in GenBank.

## Antimicrobial susceptibility testing

Antimicrobial susceptibilities were determined by Kirby-Bauer disk diffusion method recommended by the Clinical and Laboratory Standards Institute (*CLSI, 2016*). Eleven antibiotics used in clinical and veterinary medicine were tested: ampicillin 10 μg (AMP), vancomycin 30 μg (VAN), erythromycin 15 μg (ERY), tetracycline 30 μg (TET), ciprofloxacin 5 μg (CIP), norfloxacin 10 μg (NOR), nitrofurantoin 300 μg (NIT), rifampicin 5 μg (RIF), chloramphenicol 30 μg (CHL), gentamicin 120 μg (GEN) and streptomycin 300 μg (STR).

Intermediate and resistant strains were considered in a single category and classified as resistant. Strains showing resistance to three or more unrelated antibiotics were considered as multidrug-resistant (MDR) (*Schwarz et al., 2010*).

## Detection of resistance and virulence genes

Erythromycin-resistant strains were tested by PCR for the presence of resistance encoding genes more commonly associated to clinical and environmental enterococci: *erm* (B), which encodes a ribosomal methylase that mediates MLSB resistance; and *msr* C, which encodes for a macrolide and streptogtamin B efflux pump. The presence of virulence associated genes *gelE* (gelatinase enzyme), *cylA* (activator of cytolysin), *ace* (accessory colonization factor), *esp* (associated to biofilm formation) and *agg* (aggregation substance) was determined by PCR in all enterococcal isolates. The amplifications were performed as described in *Prichula et al. (2016)*. Amplifications were prepared as described to *tuf* gene. PCR amplification was performed in the conventional thermocycler (Applied Biosystems 2720 Thermal Cycler) according to the following program: 94 °C for 3 min followed by 35 cycles of 94 °C for 1 min, appropriate annealing temperature for each resistance or virulence gene for 1 min, extension at 72 °C for 1 min, followed by final extension at 72 °C for 5 min. The DNA fragment amplified by PCR was analyzed in 1.5% (w/v) agarose gels stained with SYBR® Safe DNA Gel, and visualized on a photocumenter. The sequences of the primers and annealing temperature are described in Table 1.

## Molecular typing of Enterococci by *Pulsed-field gel electrophoresis* (PFGE)

Eight six enterococci strains isolated from siblings and non-sibling caterpillars were selected for PFGE analysis according to the following criteria: maternal origin (females HEAB2, HEV2 or HES2), hatched larvae, enterococcal species and antimicrobial profile. Chromosomal DNA extraction and electrophoresis conditions were prepared according to *Murray et al. (1990)* and *Saeedi et al. (2002)*. The restriction enzyme used was *Sma* I (Invitrogen®). The electrophoresis was carried out using a clamped homogeneous electric field (CHEF-DRII device; Bio-Rad Laboratories, Richmond, Calif.), with ramped pulse times recommended by *Saeedi et al. (2002)* at 11 °C. Lambda Ladder PFG Marker (New England Biolabs) was used. The gels were stained with ethidium bromide (0.5 µg/mL for 20 min). The PFGE patterns were interpreted using the program GelCompar II v. 11 6.6, with 1.0% of tolerance, and the percentage of similarity was estimated using the Dice coefficient. The pulsotypes were clustered using the unweighted pair group whit arithmetic averages (UPGMA). A dendrogram was generated to examine the relatedness of PFGE patterns for selected isolates, and cutoff level of 80% applied to this dendrogram (*Tenover et al., 1995*).

## Statistical Analysis

Simpson's index of diversity (D) was calculated to assess the differentiation of enterococci species among the caterpillars from different maternal origins (*Hunter & Gaston, 1988*).

# RESULTS

## Enterococci species present in fecal samples of *Heliconius erato phyllis caterpillars*

A total of 178 strains of *Enterococcus* were isolated from fecal samples from fifth-instar caterpillars (Table 2). *Enterococcus casseliflavus* was the most common species identified (74.15%; $n = 132$), followed by *E. mundtii* (21.34%; $n = 38$) and *E. faecalis* (1.12%; $n = 2$). Six strains (3.37%) could not be identified to species level.

Differences in the composition of enterococci were detected between the three groups of caterpillars, as shown in Table 2. The Simpson's diversity index was different between the three populations, with higher diversity of enterococci species from fecal samples of caterpillars from HES2 ($1–D = 0.68$), followed by HEV2 ($1–D = 0.49$) and HEAB2 ($1–D = 0.27$).

## Antimicrobial susceptibility

One hundred and twenty (67.41%) enterococci were resistant to at least one evaluated antimicrobial agent. The frequency of antibiotic-susceptible strains is shown in Table 3. The rifampicin-resistance phenotype was the most commonly observed (56%; $n = 100$), followed by erythromycin (31%; $n = 55$). Eight strains (4%) were resistant to norfloxacin and five (3%) to ciprofloxacin. All investigated strains were susceptible to ampicillin, vancomycin, tetracycline, nitrofurantoin, chloramphenicol, gentamicin, and streptomycin.

Single- (SR), double- (DR), and multiple-drug resistance (MDR) were observed in 67% ($n = 80$), 28% ($n = 34$), and 5% ($n = 6$) of strains, respectively, of the 120 resistant strains.

**Table 2** Distribution of *Enterococcus* sp. isolated from fecal samples of *Heliconius erato phyllis* caterpillars.

| Species | Number (%) of enterococci strains isolated of caterpillars from[a]: | | | |
|---|---|---|---|---|
| | HEAB2 | HEV2 | HES2 | Total (%) |
| *E. faecalis* | 2 (2.04) | 0 | 0 | **2 (1.12)** |
| *E. casseliflavus* | 83 (84.69) | 42 (65.62) | 7 (43.75) | **132 (74.15)** |
| *E. mundtii* | 13 (13.26) | 19 (29.68) | 6 (37.50) | **38 (21.34)** |
| *Enterococcus* sp. | 0 | 3 (4.68) | 3 (18.75) | **6 (3.37)** |
| **Total** | **98 (100)** | **64 (100)** | **16 (100)** | **178 (100)** |

Notes.
[a] HEAB2, female from Águas Belas; HEV2, female from Viamão; HES2, female from São Francisco de Paula.

**Table 3** Antibiotic resistance profiles in enterococci isolated from fecal samples of *Heliconius erato phyllis* caterpillars.

| Female[a] | Species (n) | Number (%) of resistant strains[b] | | | | Profiles[c] | | |
|---|---|---|---|---|---|---|---|---|
| | | ERY | CIP | NOR | RIF | SR | DR | MDR |
| HEAB2 | *E. faecalis* (2) | 2 (100) | 0 | 0 | 2 (100) | 0 | 2 (100) | 0 |
| | *E. casseliflavus* (83) | 25 (30) | 4 (5) | 8 (10) | 70 (84) | 48 (58) | 21 (25) | 5 (6) |
| | *E. mundtii* (13) | 0 | 0 | 0 | 0 | 0 | 0 | 0 |
| HEV2 | *E. casseliflavus* (42) | 27 (64) | 0 | 0 | 23 (55) | 28 (67) | 11 (26) | 0 |
| | *E. mundtii* (19) | 0 | 0 | 0 | 0 | 0 | 0 | 0 |
| | *Enterococcus* sp. (3) | 0 | 0 | 0 | 3 (100) | 3 (100) | 0 | 0 |
| HES2 | *E. casseliflavus* (7) | 0 | 0 | 0 | 0 | 0 | 0 | 0 |
| | *E. mundtii* (6) | 0 | 0 | 0 | 0 | 0 | 0 | 0 |
| | *Enterococcus* sp. (3) | 1 (33) | 1 (33) | 0 | 2 (67) | 1 (33) | 0 | 1 (33) |
| **Total (178)** | | **55 (31)** | **5 (3)** | **8 (4)** | **100 (56)** | **80 (45)** | **34 (19)** | **6 (3)** |

Notes.
[a] HEAB2, female from Águas Belas; HEV2, female from Viamão; HES2, female from São Francisco de Paula.
[b] Antibiotics: ERY, erythromycin; CIP, ciprofloxacin; NOR, norfloxacin; RIF, rifampicin.
[c] Profiles: SR, single resistant; DR, double resistant; MDR, multidrug resistant.

## Determinates of resistance and virulence

None of the 55 erythromycin-resistant strains was positive for *erm*(B) and *msr*C genes. The presence of virulence genes was evaluated in all strains, and the *esp* gene was detected in 35.39% ($n = 63$), *ace* in 6.74% ($n = 12$) and *gel*E in 1.12% ($n = 2$). No strain was positive for the *cly*A and *agg* genes.

## Genetic relationships between enterococci isolated from sibling and non-sibling caterpillars in the fifth-instar

Of the 178 strains isolated, 86 (*E. casseliflavus*, $n = 58$; *E. mundtii*, $n = 23$; *E. faecalis*, $n = 2$; and *Enterococcus* sp., $n = 3$) were chosen for PFGE (Table S1). From the sibling caterpillars, numbered 6, 7, 10, 11 and 14, of the HEAB2 female were picked *E. casseliflavus* ($n = 32$), *E. mundtii* ($n = 8$) and *E. faecalis* ($n = 2$). All strains were isolated from caterpillars hatched closely in time and fed the same batch of *P. suberosa* leaves. From the offspring of the HEV2 female (sibling caterpillars 9, 18, 26, 27, and 29) *E. casseliflavus* ($n = 19$), *E. mundtii* ($n = 9$) and *Enterococcus* sp. ($n = 3$) were selected. These isolates were recovered from siblings that hatched at different times and were fed different batches of leaves. From the offspring

of the HES2 female (sibling caterpillars 3 and 17) *E. casseliflavus* ($n = 7$) and *E. mundtii* ($n = 6$) were selected.

The hierarchical relationship between enterococci selected from sibling and non-sibling caterpillars showed 22 patterns (15 patterns and 7 single strain–[singleton]; Fig. 2). Four patterns generated by PFGE indicated a genetic relationship between strains isolated from sibling caterpillars (P7, P8, P9, and P13); 11 were composed of strains isolated from the same caterpillars (P1, P2, P3, P4, P5, P6, P10, P11, P12, P14, and P15). No genetic relationships were observed for strains isolated from non-siblings.

The band patterns for *E. casseliflavus* ($n = 32$) isolates from sibling caterpillars (6, 7, 10 11 and 14) of the HEAB2 female showed six PFGE patterns (P5, P7, P8, P9, P11 and P12) and three singletons. Three PFGE patterns (P7, P8, and P9) included 18 of the 32 strains that were isolated from sibling caterpillars 6, 7, 10, and/or 11, with low levels of genetic variability. P5 and P11 each contained two isolates, and P12 with eight isolates showed genetic variation; the remaining three *E. casseliflavus* isolates were singletons and represented unique PFGE patterns. All *E. mundtii* isolates from caterpillar 14 were genetically closely related and were clustered into one pattern (P10), as were the two *E. faecalis* (P3) isolates from caterpillar 6. These results demonstrate that strains may originate from a single lineage.

Seven different PFGE patterns (P4, P6, P13, P15, and three singletons) were obtained from sibling caterpillars 9, 18, 26, 27, and 29 of the HEV2 female. The analysis of the fragment profiles of the *E. mundtii* ($n = 9$) and *Enterococcus* sp. ($n = 1$) strains isolated from caterpillars 18, 26, 27, and 29 demonstrated a genetic relationship between them (P13). The 19 *E. casseliflavus* strains showed four PFGE patterns: P4, P6, and P15, each containing six, three, and nine isolates, respectively, and one singleton. These distinct patterns are suggestive of genetic events in these strains.

The seven *E. casseliflavus* strains from caterpillars 3 and 17 (offspring of the HES2 female) showed two distinct patterns (P1 and P2) with 100% of genetic similarity between them. P1 contained four isolates and P2 had three isolates. Of the six *E. mundtii* isolates from caterpillar 3, five showed 100% similarity and were clustered in the P14 pattern, suggesting that these strains may be progeny from a single lineage. One strain had distinct and unrelated PFGE by the criteria of Tenover et al. (1995). In addition, most of the patterns were shared by isolates with the same antimicrobial profile.

## DISCUSSION

An increasing number of studies have aimed to investigate the microbial communities in the GI tract of insects. In the Lepidoptera, a high abundance of *Enterococcus* has been found, both in immatures and adults, raising questions about the role of these bacteria in invertebrates and their importance in maintaining the health of individuals (*Hammer, McMillan & Fierer, 2014*; *Holt et al., 2015*; *Chen et al., 2016*; *Snyman et al., 2016*; *Shao et al., 2017*; *Van Schooten et al., 2018*; *Allonsius et al., 2019*). To our knowledge, only *Hammer, McMillan & Fierer (2014)* and, more recently, *Van Schooten et al. (2018)* have addressed the microbial communities in the *Heliconius* GI tract, and have reported that the genus *Enterococcus* is dominant in microbial samples from different *Heliconius* species.

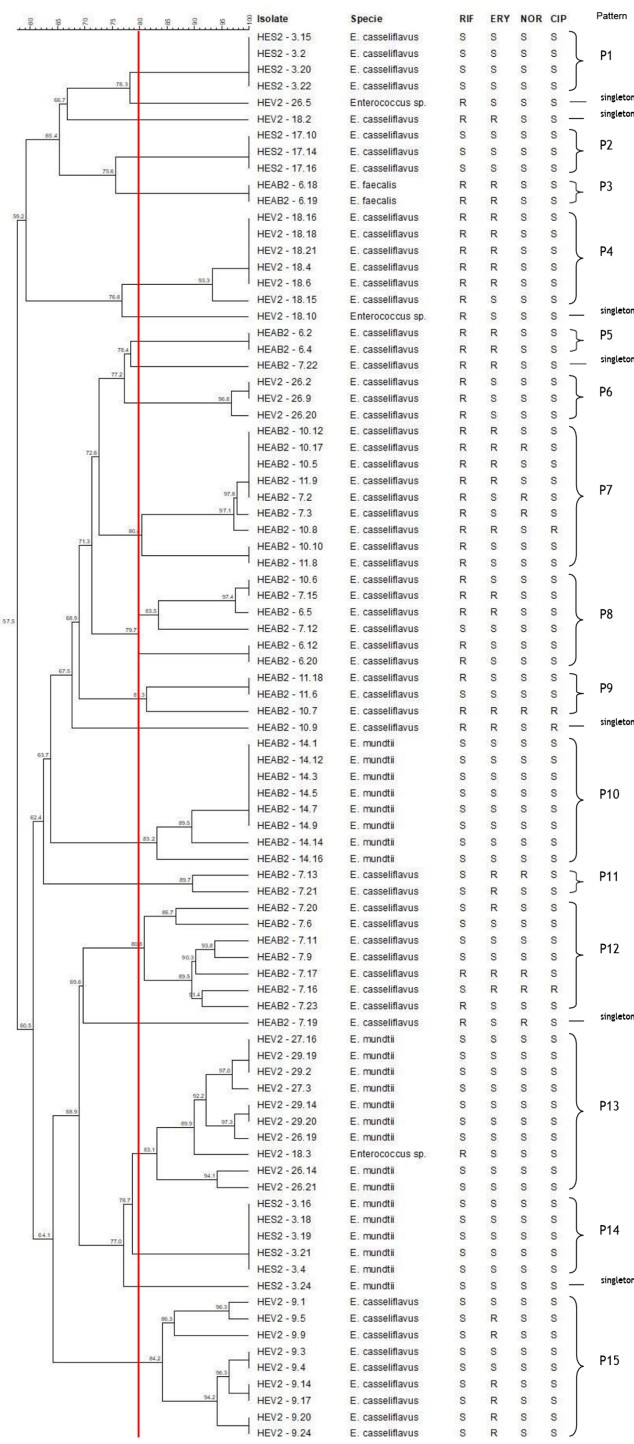

**Figure 2  Dendrogram of enterococci isolated from fecal samples of *Heliconius erato phyllis* caterpillars.** HEAB2, *H. erato phyllis* from Águas Belas; HEV2, *H erato phyllis* from Viamão; HES2, *H. erato phyllis* from São Francisco de Paula. Antibiotics: RIF, rifampicin; ERY, erythromycin; NOR, norfloxacin; CIP, ciprofloxacin; S, susceptible; R, resistant.

*Enterococcus* is associated with the environment and a wide range of organisms, including plants (*Müller et al., 2001*; *Byappanahalli et al., 2012*; *Sánchez Valenzuela et al., 2012*). In the present study, the most abundant *Enterococcus* species in the feces of *H. erato phyllis* caterpillars was *E. casseliflavus*. The diet of insects constitutes an additional source of microorganisms in the GI tract. Enterococci in the GI tract of caterpillars can play an important role in the protection against other pathogens, since these bacteria are able to produce lactic acid (causing a decrease in pH) and enterocins (peptides with antimicrobial activity). Since *E. casseliflavus* is frequently a part of microbial communities on plants (*Byappanahalli et al., 2012*; *Micallef et al., 2013*; *Ong et al., 2014*), the predominance of this species in fecal samples of *H. erato phyllis* caterpillars could be explained by the plant diet of juveniles, resulting in the dominance of this species. As noted by *Chen et al. (2016)*, enterococci stably colonize the larval gut of *Spodoptera littoralis* (Lepidoptera: Noctuidae), and may act beneficially in caterpillars by promoting nutrient supplementation (metabolism of terpenoids and polyketides) and host defense (production of antimicrobials by *E. mundtii*). However, further studies are needed to understand the role of these bacteria as members of the GI tract in *H. erato phyllis* caterpillars.

Resistant enterococcal strains isolated from fecal samples of caterpillars are a matter of concern, since these larvae had not been exposed to antibiotics. Studies that highlight the resistance profile of enterococci isolated from insects in natural environments are scarce. *Channaiah et al. (2010)* described enterococci isolated from insect pests of stored products; they were resistant to tetracycline, streptomycin, erythromycin, kanamycin, ciprofloxacin, ampicillin, and chloramphenicol; this suggests that these animals can be potential vectors in disseminating antibiotic-resistant strains. *Ahmad et al. (2011)* reported MDR enterococci isolated from house flies and cockroaches in a confined swine production environment, and have suggested that these insects may be potential vectors and/or reservoirs of resistant enterococci. Despite the occurrence and spread of resistant strains intensifying due to the use of antimicrobials, the isolation of resistant enterococci in the present study could be related to the resistance that occurs naturally in the environment (environmental resistome) and/or to anthropogenic activities (contamination of the environment) (*Martínez, 2008*; *Allen et al., 2010*).

The most widespread mechanism of resistance to macrolides in enterococci is mediated by the *erm* and *msr* C genes (*Aarestrup et al., 2000*; *Santestevan et al., 2015*; *Prichula et al., 2016*). Nevertheless, none of these genes was detected in the present study. It is possible that these strains harbored other erythromycin-resistance genes, such as *erm* D, E, and F, and other efflux pump genes such as *msr* A. A low percentage of virulence genes was detected in enterococci of *H. erato phyllis* caterpillars. Although these genes are related to pathogenicity of clinical enterococcal strains, their presence in strains in fecal samples from caterpillars may be associated with the maintenance of cells of the GI tract, and consequently with microorganism and host interactions.

From the analysis of the PFGE fingerprint, *E. casseliflavus*, *E. faecalis*, *E. mundtii*, and *Enterococcus* sp. isolated from fecal samples of sibling and non-sibling caterpillars demonstrated unrelated or related patterns based on maternal origin. The unrelated patterns found in P1, P2, P3, P4, P5, P6, P10, P11, P12, P14, and P15 demonstrated genetic

diversity among these strains. The genetic variation in these strains may be associated with genetic events, such as mobile elements or mutation, a common characteristic of enterococci (*Lebreton, Willems & Gilmore, 2014*). The related patterns observed among strains isolated from sibling caterpillars (P7, P8, P9, and P13) may be associated with a common source, e.g., diet (herbivory) and/or vertical transmission (through the egg surface). Since the fecal samples used in the present study were collected from fifth-instar caterpillars, the last stage before the pupa, the results present here may be suggest vertical transmission of enterococci that are being replaced from the diet. Plants are a food source for bacteria present in the GI tract of insects; these bacteria improve the quality of diets poor in nutrients and take part in development and maturation of the immune system to protect the host against pathogenic microorganisms (*Dillon & Dillon, 2004*; *Engel & Moran, 2013*). Therefore, it is likely that herbivory provides an abundant supply of enterococci throughout the larval stage of *H. erato phyllis*. Considering that *Passiflora* leaves are the only food of the caterpillars, those leaves could be the source of *Enterococcus* sp. in their GI tracts.

Besides the diet, vertical transmission from the female to her offspring can also be a source of bacteria. Some studies have described the mechanism for vertical transmission of bacteria in different species of Lepidoptera (*Brinkmann, Martens & Tebbe, 2008*; *Chen et al., 2016*; *Teh et al., 2016*; *Shao et al., 2017*). The caterpillar hatches by chewing a hole in the chorion and emerging through it from the egg. *Brinkmann, Martens & Tebbe (2008)* reported that the *Enterococcus* spp. present in the gut of the larvae of *Manduca sexta* (Lepidoptera: Sphingidae) were acquired via ingestion of their eggshell, demonstrating maternal transmission of microorganisms. *Chen et al. (2016)* analyzed the composition and activity of microbiota in the moth *Spodoptera littoralis*, which is an agricultural pest; they found that enterococci associated with adult females were also in the egg mass, and further colonized the larval gut of individuals, suggesting vertical transmission of these bacteria. *Teh et al. (2016)* showed the route of transmission of *E. mundtii* in *S. littoralis* when administered *in vivo*; the authors reported the presence of *E. mundtii* at all life stages of this insect. In addition, they established the presence of these bacteria in oocytes and the muscle tissue in the first-instar larvae of the second-generation offspring, highlighting again the vertical transmission of enterococci in this lepidopteran. The enterococci isolated in the present study may be linked to herbivory; however, although our analysis does not include adult females and the bacterial communities of *Passiflora* leaves, we do not rule out the possibility that enterococci could also be transmitted from the female to the offspring through the surface of the egg, as previously demonstrated for other species of Lepidoptera.

## CONCLUSIONS

*Enterococcus casseliflavus* was the dominant enterococcal species isolated in fecal samples of the fifth-instar caterpillars of *H. erato phyllis*. Resistant strains present in the caterpillars studied may be related to the environmental resistome and/or anthropogenic activity. In addition, the presence of MDR enterococci may be an indication of contamination

of the environmental by antibiotics. The results obtained by PFGE analysis suggest that the enterococci isolated from fecal samples of sibling caterpillars might have come from common sources, e.g., the diet (herbivory) and/or vertical transmission (through the egg surface). Further studies will be conducted to better understand the role of *Enterococcus* in the GI tract microbial community of *H. erato phyllis* butterflies, and the mechanisms involved in acquisition and maintenance of these bacteria. In addition, the data obtained can be used in future comparative analyses of the microbiota present in adult *H. erato phyllis* females and their offspring, to confirm the occurrence of vertical transmission of *Enterococcus* sp. in this model organism.

## ACKNOWLEDGEMENTS

We thank Lucas de Oliveira Einsfeld and Gabriella Oliveira de Araújo for the drawing.

### Funding

This work was supported by the Conselho Nacional de Desenvolvimento Científico e Tecnológico do Brasil, CNPq - #302574/2017-4, #401714/2016-0 and #303603/2015-1 and Coordenação de Aperfeiçoamento de Pessoal de Nível Superior (CAPES). The funders had no role in study design, data collection and analysis, decision to publish, or preparation of the manuscript.

### Grant Disclosures

The following grant information was disclosed by the authors:
Conselho Nacional de Desenvolvimento Científico e Tecnológico do Brasil.
CNPq: #302574/2017-4, #401714/2016-0 and #303603/2015-1.
Coordenação de Aperfeiçoamento de Pessoal de Nível Superior (CAPES).

### Competing Interests

The authors declare there are no competing interests.

### Author Contributions

- Rosana Huff conceived and designed the experiments, performed the experiments, analyzed the data, prepared figures and/or tables, authored or reviewed drafts of the paper, and approved the final draft.
- Rebeca Inhoque Pereira conceived and designed the experiments, performed the experiments, analyzed the data, authored or reviewed drafts of the paper, and approved the final draft.
- Caroline Pissetti performed the experiments, analyzed the data, prepared figures and/or tables, authored or reviewed drafts of the paper, and approved the final draft.
- Aldo Mellender de Araújo conceived and designed the experiments, analyzed the data, authored or reviewed drafts of the paper, and approved the final draft.
- Pedro Alves d'Azevedo and Jeverson Frazzon analyzed the data, authored or reviewed drafts of the paper, and approved the final draft.

- Ana Paula Guedes Frazzon conceived and designed the experiments, analyzed the data, prepared figures and/or tables, authored or reviewed drafts of the paper, and approved the final draft.

## Field Study Permissions

The following information was supplied relating to field study approvals (i.e., approving body and any reference numbers):

This study was carried out in accordance with the recommendations of Chico Mendes Institute for Biodiversity Conservation (ICMBio).

The protocol was approved by Information Authorization System in Biodiversity (SISBIO) number 33404-1. This study has the approval of the Council for the Management of Genetic Patrimony - CGEN - under the Ministry of Environment number A720680.

## Data Availability

The raw data and eletrophoresis gels are available in the Supplemental Files.

## Supplemental Information

Supplemental information for this article can be found online at http://dx.doi.org/10.7717/peerj.8647#supplemental-information.

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
