# Peer review of "Antimicrobial resistance and genetic relationships of enterococci from siblings and non-siblings Heliconius erato phyllis caterpillars"

_PeerJ, doi:10.7717/peerj.8647_

## Round 0.1 · original submission · Minor Revisions

Dear Dr. Huff and colleagues:

Thanks for submitting your manuscript to PeerJ. I have now received three independent reviews of your work, and as you will see, the reviewers raised some concerns about the research. Despite this, these reviewers are optimistic about your work and the potential impact it will have on research communities studying enterococci associated with Lepidoptera. Thus, I encourage you to revise your manuscript, accordingly, taking into account all of the concerns raised by both reviewers.

The reviewers primarily raised concerns regarding English usage, overall writing and methods associated with tree-building. Please address all of these issues, and perhaps enlist the help of a native English speaker before resubmitting your manuscript.

Good luck with your revision,

-joe

Reviewer 1 ·

Basic reporting

see entire review below

Experimental design

see entire review below

Validity of the findings

Manuscript. No.: #41227

Title: Antimicrobial resistance and genetic relationship among enterococci from sibling and non-siblings Heliconius erato phyllis caterpillars

Authors: Rosana Huff, Rebeca Inhoque Pereira, Caroline Pissetti, Aldo Mellender de Araújo, Pedro Alves d’Azevedo, Jeverson Frazzon, Ana Paula Guedes Frazzon


Overall, the manuscript is well written- clear, professional and scientific English language is used throughout. The study’s experimental design needs more work. Specifically, the authors will need to provide more information on a) why only one antibiotic resistance gene was tested, and b)why their results don’t correlate between the susceptibility testing results (particularly erythromycin resistance) and the detection of the erythromycin resistant gene ermB (see detailed comment below under results section of this review).

Please see below for my specific comments to improve the manuscript.

Abstract and Introduction:
The introduction needs more detail and background with relevant citations to establish a) why studying the antimicrobial resistance (AMR) and multidrug resistance (MDR) is overall so important considering the current global AMR crisis, and b) why studying AMR and MDR is particularly important in insects. Providing more background information on our current knowledge regarding the role of insects in the dissemination of AMR/MDR would also be very beneficial for the readers of the journal.

Line 26 – bacteria instead of bacterial.
Line 41 – instead of ‘submitted to’, use ‘analyzed for’
Line 79 – please elaborate on what ‘host’ you’re indicating here

Materials and methods:
Lines 128-129 – how are these eggs transported?
Line 130 – How many fecal samples were collected? I see you mention this in your abstract as n=12, you should also put this in the actual methods section

Lines 153 – 156 – the DNA extraction method needs more detail, only citing a paper isn’t enough. Was a kit used? What was the elution volume? Did you measure DNA concentrations? PCR methods also need to be far more detailed. What is the mastermix composition? What is the program used? Where were the mastermix components purchased from?

Line 154 – tuf gene – at first use, please explain what the function of this gene is. This helps readers with adequate information for easy understanding.
Lines 161 – 162 – PCR assay need more detail. I see you provide table 1, but this does not have the mastermix compositions, the thermal cycler programs. Please include this information either in the text or in the table as separate columns.

Lines 173 – 182 – Antimicrobial susceptibility testing – Please detail on the actual protocol used. Just a reference is not enough as this is one of the most important methods you have used for generating your dataset. What standard was used to assess whether the strains were susceptible, intermediate or resistant?

Lines 184 – 187 – Why was only one antibiotic resistance gene chosen? I see you tested for so many other antibiotics for your Kirby Bauer assay, but you have tested only one of the genes. There are hundreds of genes that could be tested not just for presence, as done in this study, but also for quantification of these genes in the starting sample collected. Therefore, selecting only one gene for detection does not make much impact as per a study like this is concerned.

Lines 184 – 191 – The PCR methods need more detail in this section as well as mentioned above.

RESULTS:
Antimicrobial Susceptibility – lines 225 – 233 – Instead of a table, I recommend the authors provide a figure of this for better understanding. You could divide your plot into the different strains of Enterococci tested, and make a stacked plot with the results from the different antibiotics tested.

Lines 235 – 236 –
• Why only one antibiotic was tested for the gene presence need to be explained.
• Authors also need to explain why the 55 strains tested negative for ermB, and all 55 strains tested positively resistant for erythromycin as per their antimicrobial susceptibility testing results.

Line 244 – I did not notice a table S1 in the submitted manuscript packet

Reviewer 2 ·

Basic reporting

The article is generally well structured. Enough background information is provided to stress the importance of the topic. I have certain concerns considering English language that will be listed below in the field General comments for the author. Although I made specific comments and suggested corrections, a native speaker should check the manuscript. The figures and tables are appropriate, corresponding to the text well – I have made minor objections only.

Experimental design

The study matches aims and scopes of the Journal. The aims are clear and are logically related to the background information. The research question is meaningful and relevant in light of scarce data about the topic.

Validity of the findings

The results are valid. Their importance is unquestionable in light of serious problem of growing antibiotic resistance. Conclusions are clear, logic and based on the results.

Additional comments

Abstract

Line 26-27: In the sentence “Studies evaluating bacterial in insects could provide information about host-microorganism-environment interactions.” What do you mean by “bacterial”? Is it bacterial community, bacteria, microbiome or something similar? “Bacterial” is an adjective.
Lines 53-54: In the sentence “Resistant enterococci strains could be related to environmental pollution or linked to environmental resistome.” It would be better to erase “linked to”.
Lines 54-57: In the sentence “The PFGE analysis showed related genetic relationship among some strains, suggesting that the enterococci isolated from fecal samples of fifth instar sibling caterpillars might have come from common sources, by diet (herbivory) and/or via vertical transmission (through egg surface).…” I believe that the word “related” is redundant. If there is a relationship between the two, then it is obvious that those are related. In addition, I would write “…via diet (herbivory) or vertical transmission (through egg surface)”.
Lines 57-59: In the sentence “Further studies will be conducted to better understand the role of Enterococcus on the microbial gastrointestinal tract community of these insects…” I believe there should be “…in the microbial gastrointestinal tract community of these insects…”

Introduction

Line 62: please improve English. “Heliconius (Lepidoptera; Nymphalidae) comprises numerous widespread species of butterflies” or Heliconius is or represents a widespread genus of butterflies”. To comprise means to contain. So, genus can only contain species.
Line 80: please rewrite. I don’t understand “Gut microbial community provide an especially diverse range of benefits in insect nutrition, e.g. by providing xenobiotic metabolism? To provide means to give someone something that they need (Cambridge Dictionary). So, how is the gut microbial community related to xenobiotic metabolism”?
Line 83: to introduce an abbreviation „GI tract“, you should provide full term here when first mentioned, not in the line 85. In the line 85 the abbreviation is enough.
Lines 86-88: It is hard to understand this sentence. In addition, there should stand “one of the most frequent genera” instead of “the most frequently genus”. Anyway, the whole sentence should be rewritten. For example: Enterococcus is one of the most frequent bacterial genera present in the gut microbiota at different life stages of Lepidoptera (Brinkmann, Matens & Tebbe, 2008; Chen et al., 2016; Ruokolainen et al., 2016; Snyman et al., 2016; van Shooten et al., 2018; Allonsius et al., 2019).
Lines 92-94: I would write like this: Hammer et al. (2014) reported that Enterococcus was the most abundant genus found in immature stages and adults of H. erato from Panamá. First, if it is Enterococcus spp, as you wrote, then it is related to species of that genus. Second, Hammer et al. (2014) did not mention subspecies of H. erato, as I could see in the original article. If I am wrong, then please name that subspecies you referred to.
Line 105: Previusly was not mentioned that Enterococcus might play a fundamental role in the development of those insects. Perhaps you intended to state that implicitly earlier in the text, but here it sounds too explicitly without references. Please, rewrite or add references.

Results

In the Table 2 – I think it should be emphasized that it is about the strains. “Number (%) of enterococci strains isolated of caterpillars from”
In M&M (lines 118-122) you introduces three localities you collected butterflies from in this order: “The first female (HEAB2) was collected in a forest fragment located in Águas Belas Agronomical Station (30° 120 02' 18.1" S; 51° 01' 23.0" W), the second female (HES2) in a forest fragment located in São Francisco de Paula (29° 26' 34.1" S; 50° 36' 48.8" W) and the third female (HEV2) from a population in an intense urban area in Viamão (30° 09' 40.5" S; 50° 55' 01.5" W).” Then, in the lines 127-128 you change the order: “A total of 12 eggs were collected (five from HEAB2, five from HEV2 and two from HES2) with the assistance of a paintbrush. Also for example in the table 2 it is HEAB2, HEV2 and HES2. This is very confusing for a reader because the reader most probably is not familiar with the localities in your area; for example, I first remembered that the third locality was in urban area and later I noticed that I was interpreting results in a wrong way because you changed the order of localities with no reason. Please, try to be consistent throughout the manuscript as possible. And please check whole result section including tables and figures to fix this problem.

Discussion

Lines 296-297: “E. casseliflavus was more frequently identified in feces of H. erato phyllis caterpillars”. Compared to what?
Line 297: Perhaps it is better to write “The diet constitutes an additional source of microorganisms”
Line 299: with regard to; “since these bacteria are able to…” instead of “since this genus are…”
Line 301-304: Unclear sentence; I would put it like this: “Since E. casseliflavus is frequently a part of microbial communities in plants (Byappanahalli et al., 2012; Micallef et al., 2013; Ong et al., 2014), the predominance of this species in fecal samples of H. erato phyllis caterpillars could be explained by the herbivory of juveniles.” I do not understand what it means “unleashing the predominance of these species”.
Lines 304-305: Please explain in more detail how enterococci may act beneficially for caterpillars.
Line 319: Delete “linked to”
Line 326: Delete coma after “Although” and before “may be associated”
Lines 341-342: “these bacteria improve quality of diets poor in nutrients and take part in development…”
Lines 344-345: …herbivory enables abundant supply…
Line 346-347: …those leaves could be the source…
Line 348: Besides the diet, the vertical transmission…
Lines 354-360: This is too complicated, it should be rewritten.
Lines 361-362: Teh et al. (2016) showed the route of transmission of E. mundtii in S.littoralis when administered in vivo; the authors reported the presence of E. mundtii at all life stages of this insect. In addition, they established the presence of this bacteria in oocytes and the muscle tissue in the first-instar larvae of the second-generation offspring, highlighting again the vertical transmission of enterococci in this butterfly species.
Line 365: despite our analysis does not include
Line 366: we do not discard the possibility that enterococci…

Conclusions

Line 371: E. casseliflavus was the dominate genus
Line 374: delete “also”
Line 376-377: via diet (herbivory) and/or vertical transmission (through egg surface).
Line 378: in the microbial GI tract

Reviewer 3 ·

Basic reporting

On line 170-171, please elaborate on the quality of the sequences generated. Please mention: how many base pairs were the sequences? Did you use any filtering criteria? If so, how many sequences did you have initially and how many did you end up wit after filtering.

On line 204-206: What parameters were used to construct this dendogram? For example, did you construct a maximum likelihood tree? Describe again how many sequences you used. Also describe, what was the outgroup used for the tree. How should the branch lengths of the tree be interpreted. Right now the methods in this section seem very vague and they need more details.

Experimental design

No comments

Validity of the findings

No comments

Additional comments

I really enjoyed reading this paper. I think this paper adds an additional dimension to microbiome research in Lepidoptera. The missing details mentioned above should be added before Acceptance.

---

## Round 0.2 · accepted · Accept

Dear Dr. Huff and colleagues:

Thanks for re-submitting your revised manuscript to PeerJ, and for addressing the concerns raised by the reviewers. I now believe that your manuscript is suitable for publication. Congratulations! I look forward to seeing this work in print, and I anticipate it being an important resource for research communities studying enterococci associated with Lepidoptera.

Thanks again for choosing PeerJ to publish such important work.

-joe

Reviewer 1 ·

Basic reporting

The authors in their "Rebuttal letter" have answered the questions and comments I had previously made on the earlier version of the manuscript. The authors, however, do mention in the letter on few instances (some of the methods and results/figures comments) that they did not make some of the changes to the manuscript as they thought it was not necessary.

I still do think that making those changes/ adding those elements into the main manuscript will enhance the quality of the manuscript as a whole, and that based on PeerJ standards it is necessary to do so.

Experimental design

see comment above

Validity of the findings

see comment above

Reviewer 3 ·

Basic reporting

The article is revised well and the english is clear now.

Experimental design

The authors have done a great job in revising the methods and they are more articulate now.

Validity of the findings

Overall the paper looks great now after all the revisions.